# Novel Noise Injection Scheme to Guarantee Zero Secrecy Outage under Imperfect CSI

**DOI:** 10.3390/e25121594

**Published:** 2023-11-28

**Authors:** Hien Q. Ta, Lam Cao, Hoon Oh

**Affiliations:** 1School of Electrical Engineering, International University, Ho Chi Minh City 700000, Vietnam; tqhien@hcmiu.edu.vn (H.Q.T.); lamct25@mp.hcmiu.edu.vn (L.C.); 2Vietnam National University, Linh Trung Ward, Thu Duc District, Ho Chi Minh City 700000, Vietnam; 3Department of Electrical, Electronic and Computer Engineering, University of Ulsan, Ulsan 44610, Republic of Korea

**Keywords:** physical layer security, artificial noise, zero secrecy outage

## Abstract

The paper proposes a novel artificial noise (AN) injection strategy in multiple-input single-output multiple-antenna-eavesdropper (MISOME) systems under imperfect channel estimation at the legitimate channel to achieve zero secrecy outage probability under any circumstance. The zero secrecy outage is proved to always be achievable regardless of the eavesdropper’s number of antennas or location when the pair secrecy and codeword rates are chosen properly. The results show that when there is perfect channel state information, the zero-outage secrecy throughput increases with the transmit power, which is important for secrecy design. Additionally, an analysis of the secrecy throughput and secrecy energy efficiency gives further insight into the effectiveness of the proposed scheme.

## 1. Introduction

Wireless communication has become an indispensable part of the modern world, enabling the connection and exchange of information between various devices and networks. As wireless communication continues to develop, the demand for more sophisticated security mechanisms grows, seeking to further protect sensitive data and ensure the privacy and integrity of communication. The demand for security in wireless communication arises from several factors. Firstly, wireless networks are by nature more vulnerable to security threats such as eavesdropping, interception, and unauthorized access compared to wired networks. This makes it essential to employ robust security mechanisms to safeguard transmitted data. Additionally, the continuous advancement and widespread adoption of wireless technologies have expanded the attack surface for potential threats. Traditionally, encryption techniques such as advanced encryption standard (AES) [1] and RSA [2] have been employed to encrypt data packets, making them unintelligible to unauthorized recipients. Encryption safeguards the confidentiality of transmitted data and prevents unauthorized tampering or modification during transmission. However, data encryption has certain disadvantages, such as requiring additional resources and associated setup and maintenance, increasing complexity, and reducing data processing speed.

In the past decades, physical layer security (PLS) has arisen as a propitious solution to improve the overall protection of wireless networks. The demand for PLS stems from the fact that cryptographic methods alone may not be sufficient to address all security challenges in wireless communications. PLS techniques take advantage of various physical properties, such as signal strength, fading effects, noise, and channel characteristics, to establish secure communication links. Wyner laid the foundation for physical layer security (PLS) through the study of the wiretap channel [3]. Building upon this concept, subsequent research focused on studying the secrecy in Gaussian wiretap channel and fading channel models. It was demonstrated in [4] that there exists a non-zero secrecy capacity even when channel conditions are more favorable for the eavesdropper.

More recently, novel approaches aimed at enhancing security have emerged. These include cooperative communication, as discussed in [5], and the introduction of artificial noise (AN) injection, as explored by [6]. Among these solutions, AN receivers have attracted significant attention, particularly in the context of comprehensive three-node systems, as shown in [7,8]. In [7], a secure half-duplex (HD) transmission scheme using AN injection is introduced. In this scheme, the receiver spends the first phase broadcasting AN, which is then processed and combined with the secret message at the transmitter during the later phase. In contrast, Ref. [8] investigates the use of a full-duplex (FD) jamming receiver, which simultaneously receives the desirable message and broadcasts AN to disrupt potential eavesdropping attempts. This approach is particularly effective when the eavesdropper and intended receiver are in close proximity, experiencing higher received AN power. The potential of FD receivers in physical layer security (PLS) has been further examined in subsequent studies, such as [9,10,11], showcasing improvements in secrecy outage probability at the cost of network connection reliability in hybrid FD/HD models with FD jamming receivers. Building on the concepts presented in [7,8], Ref. [12] proposed a secure FD transmission scheme, where the transmitter overlays the secret message with the AN coming from the receiver. Furthermore, additional research efforts have explored the use of FD relays broadcasting AN in decode-and-forward as well as amplify-and-forward relay systems, addressing scenarios with an unknown eavesdropper location [13,14]. Although PLS has been studied for years, it has limits in realistic applications, especially for a powerful eavesdropper which is located close to the transmitter or equipped with an infinite number of antennas. In fact, since the traditional AN is usually transmitted in the null space of the main channel, it will be easily canceled if the eavesdropper is powerful enough and has a large number of antennas [6]. As a result, there is no secrecy anymore, and a novel AN injection strategy is urgently required, in which the AN is transmitted in the same space as the main channel while still being canceled for legitimate receivers but not eavesdroppers by using the strategy of channel state information (CSI) leakage avoidance [15]. Furthermore, because security is required with probability 1 in some scenarios such as credit card number transmission, secrecy outage, or interception by an eavesdropper is required to be zero. As such, PLS or even other layers cannot support security in any circumstance when there is a powerful eavesdropper. Therefore, it is essential to revisit PLS for a novel design that can guarantee zero secrecy outage.

In this paper, we propose a secure transmission model using AN with a multi-antenna transmitter and an eavesdropper under imperfect channel estimation. The numerical results show an achievable zero secrecy outage probability by using a two-phase transmission protocol with channel inversion pre-coding at the transmitter. Furthermore, we also provide analysis for the secrecy throughput and in turn, the secrecy energy efficiency of the scheme.

The contributions of the paper are as follows:Different from conventional schemes, in which AN is transmitted in the null space of the secure signal, the proposed artificial noise injection scheme transmits AN in the same space as the secure signal and uses an extra time slot for only transmitting AN with aid reverse pilot training to achieve noise cancellation capability at legitimate receivers and not at an eavesdropper.Although the proposed strategy wastes one extra time slot, it reduces complexity at the transmitter, which requires one AN vector compared to an AN matrix as in the conventional strategy. This simple strategy can also be extended to apply to many other system networks to efficiently achieve zero secrecy outage.The closed-form formula of the secrecy rate subject to zero secrecy outage is determined. Then, the resulting zero-outage secrecy throughput can always be achieved regardless of how powerful the eavesdropper is, e.g., if it has an infinite number of antennas.

From here on, the paper is structured as follows: Section 2 introduces the system model of the proposed transmission scheme; Section 3 and Section 4 show the formulation of secrecy and connection outage probability, respectively; Section 5 provides analysis for the secrecy throughput and secrecy energy efficiency of the system; finally, numerical results and discussion are given in Section 6.

## 2. System Model

As depicted in Figure 1, consider a three-node secure transmission model with a transmitter (Alice) sending confidential information to a legitimate receiver (Bob), while an eavesdropper (Eve) attempts to intercept the message. Alice, Bob, and Eve have $N_{A}$, 1, and $N_{E}$ antennas, respectively. The fading channels are modeled as quasi-static Rayleigh with constant gain for each time slot that changes independently between different slots.

The proposed scheme is under two-phase transmission of a secret message plus artificial noise (AN) and only AN, in two different time slots. Considering the reverse training method, where Bob periodically transmits pilots for Alice to estimate the channel gain [16]. Assuming an imperfect minimum-mean-square-error (MMSE) receiver at Bob, the actual channel at Bob, denoted as $h_{i}$ for i∈{1,2} representing two phases of transmissions, is given by [17]
(1)$$\begin{array}{ccc} h_{i}\; & = & \;{\hat{h}}_{i}+{\tilde{h}}_{i} \end{array}$$
where ${\hat{h}}_{i}∼\mathrm{CN}\left(0,\left(1-\beta \right)\sigma_{h}^{2}I_{N_{A}\times N_{A}}\right)$ is the estimated channel gain, and ${\tilde{h}}_{i}∼\mathrm{CN}\left(0,\beta \sigma_{h}^{2}I_{N_{A}\times N_{A}}\right)$ is the error part with the estimation error coefficient denoted β. ($\mathrm{CN}(0,\sigma^{2}I_{n\times n})$ denotes the independent and identical distributed (i.i.d) complex vector where each element has Gaussian distribution with zero-mean and variance $\sigma^{2}$, and 0 and $I_{n\times m}$ denote the zero vector and identity matrix, respectively). We also assume that the background noise at each node has a zero-mean complex Gaussian distribution with variance $\sigma_{n}^{2}$.

The secure transmission has two phases:

### 2.1. Phase 1

In the first phase, Alice attempts to protect the confidential information by sending a secret message along with a superimposed AN. The secure message at Alice is transmitted with an allocated power fraction, denoted as α in the range (0,1), along with the AN as
(2)$$\begin{array}{ccc} x_{1}\; & = & \;\frac{{\hat{h}}_{1}^{*}}{∥{\hat{h}}_{1}{∥}^{2}}\left(\sqrt{\alpha }u+\sqrt{1-\alpha }v\right), \end{array}$$
where *u* and *v* denote the secret message and the AN signal, respectively, and are i.i.d zero-mean complex Gaussian with variance $\sigma_{u}^{2}$.

Due to error estimation, Bob receives
(3)$$\begin{array}{ccc} y_{b,1}\; & = & \;h_{1}^{T}x_{1}+n_{b,1}\\ \; & = & \;\left(\sqrt{\alpha }u+\sqrt{1-\alpha }v\right)+\frac{{\tilde{h}}_{1}^{T}{\hat{h}}_{1}^{*}}{∥{\hat{h}}_{1}{∥}^{2}}\left(\sqrt{\alpha }u+\sqrt{1-\alpha }v\right)+n_{b,1}, \end{array}$$
while assuming perfect CSI at Eve, it receives,
(4)$$\begin{array}{ccc} y_{e,1}\; & = & \;G_{1}x_{1}+n_{e,1}\\ \; & = & \;\frac{G_{1}{\hat{h}}_{1}^{*}}{∥{\hat{h}}_{1}{∥}^{2}}\left(\sqrt{\alpha }u+\sqrt{1-\alpha }v\right)+n_{e,1}, \end{array}$$
respectively, where $G_{1}$ denotes the channel gain at Eve in phase 1 with each element being i.i.d zero-mean complex Gaussian with variance $\sigma_{g}^{2}$. $n_{b,1}$ and $n_{e,1}$ denote the background noise at Bob and Eve, respectively.

### 2.2. Phase 2

In phase 2, only artificial noise is transmitted as
(5)$$\begin{array}{ccc} x_{2}\; & = & \;\frac{{\hat{h}}_{2}^{*}}{∥{\hat{h}}_{2}{∥}^{2}}v, \end{array}$$
which yields Bob’s and Eve’s received signals,
(6)$$\begin{array}{ccc} y_{b,2}\; & = & \;h_{2}^{T}x_{2}+n_{b,2}\\ \; & = & \;v+\frac{{\tilde{h}}_{2}^{T}{\hat{h}}_{2}^{*}}{∥{\hat{h}}_{2}{∥}^{2}}v+n_{b,2} \end{array}$$
and
(7)$$\begin{array}{ccc} y_{e,2}\; & = & \;G_{2}x_{2}+n_{e,2}\\ \; & = & \;\frac{G_{2}{\hat{h}}_{2}^{*}}{∥{\hat{h}}_{2}{∥}^{2}}v+n_{e,2}, \end{array}$$
respectively, where $G_{2}$ denotes the channel gain at Eve in phase 2, where each element is i.i.d complex Gaussian with zero-mean and variance $\sigma_{g}^{2}$. $n_{b,2}$ and $n_{e,2}$ denote the background noise at Bob and Eve, respectively.

The *total transmit power* based on the channel inversion strategy can be computed from (Equation 2) and (Equation 5) as a function of ${\hat{h}}_{1}$ and ${\hat{h}}_{2}$ [18]:(8)$$\begin{array}{ccc} P({\hat{h}}_{1},{\hat{h}}_{2})\; & = & \;E(||x_{1}{\left|\right|}^{2})+E(||x_{2}{\left|\right|}^{2})\\ \; & = & \;\sigma_{u}^{2}\left(|\right|{\hat{h}}_{1}{\left|\right|}^{-2}+\left|\right|{\hat{h}}_{2}{\left|\right|}^{-2}), \end{array}$$
which yields the average total transmit power, denoted *P*, of
(9)$$\begin{array}{ccc} P\; & = & \;E_{{\hat{h}}_{1},{\hat{h}}_{2}}\left[P({\hat{h}}_{1},{\hat{h}}_{2})\right]\\ \; & = & \;\frac{2\sigma_{u}^{2}}{\left(1-\beta \right)\sigma_{h}^{2}\left(N_{A}-1\right)}. \end{array}$$
Therefore, we obtain
(10)$$\sigma_{u}^{2}=\left(1-\beta \right)\sigma_{h}^{2}P\left(N_{A}-1\right)/2.$$

## 3. Secrecy Outage Probability

From Eve’s perspective, it is impossible to acquire $G_{2}{\hat{h}}_{2}^{*}/{∥{\hat{h}}_{2}∥}^{2}$ phase shift in phase 2 based on the received signal in (Equation 7) as it is only noise. We see that when Eve does not experience any channel noise, the phase shift of $y_{e,2}$ is determined by the phase difference between *v* and $G_{2}{\hat{h}}_{2}^{*}/{∥{\hat{h}}_{2}∥}^{2}$. Since *v* is defined as a complex Gaussian AN vector independent of the channel phase, the signals $y_{e,2}$ and $G_{2}{\hat{h}}_{2}^{*}/{∥{\hat{h}}_{2}∥}^{2}$ are also independent. As we obtain the entropy equality,
(11)$$\begin{array}{c} H\left(G_{2}{\hat{h}}_{2}^{*}/{∥{\hat{h}}_{2}∥}^{2}\right|y_{e,2})=H\left(G_{2}{\hat{h}}_{2}^{*}/{∥{\hat{h}}_{2}∥}^{2}\right), \end{array}$$
the received signal cannot give any information on the CSI, meaning that the eavesdropper can only process the secret message using (Equation 4). The SINR at Eve is given as
(12)$$\begin{array}{ccc} \gamma_{e}\; & = & \;\frac{\frac{{\left|G_{1}{\hat{h}}_{1}^{*}\right|}^{2}}{∥{\hat{h}}_{1}{∥}^{4}}\alpha \sigma_{u}^{2}}{\frac{{\left|G_{1}{\hat{h}}_{1}^{*}\right|}^{2}}{∥{\hat{h}}_{1}{∥}^{4}}\left(1-\alpha \right)\sigma_{u}^{2}+\sigma_{n}^{2}} \end{array}$$
(13)$$\begin{array}{ccc} \; & = & \;\frac{\frac{{∥g∥}^{2}}{∥{\hat{h}}_{1}{∥}^{2}}\alpha \sigma_{u}^{2}}{\frac{{∥g∥}^{2}}{∥{\hat{h}}_{1}{∥}^{2}}\left(1-\alpha \right)\sigma_{u}^{2}+\sigma_{n}^{2}} \end{array}$$
where $g:=\frac{G_{1}{\hat{h}}_{1}^{*}}{∥{\hat{h}}_{1}∥}∼\mathrm{CN}\left(0,\sigma_{g}^{2}I_{N_{e}\times N_{e}}\right)$. The capacity at Eve is given as
(14)$$\begin{array}{c} C_{e}=\frac{1}{2}\log_{2}\left(1+\gamma_{e}\right) \end{array}$$
where the factor 1/2 indicates that Eve only obtains information in the first phase.

Assume that Alice employs the wiretap code transmission scheme [3] to secure the information from Eve. The scheme uses codeword and secret rate parameters, $R_{b}$ and $R_{s}$, with $R_{b}>R_{s}$. The positive difference between $R_{b}$ and $R_{s}$ is the cost of securing the confidential information. Since ${∥g∥}^{2}∼X_{2N_{E}}\left(0,\sigma_{g}^{2}/2\right)$ ($X_{2N}\left(\sigma^{2}/2\right)$ denotes Chi-square distribution with 2N degrees of freedom and common variance $\sigma^{2}/2$), the secrecy outage probability for given transmissions is given by [19]
(15)$$\begin{array}{ccc} \;P_{so} & = & \;Pr\left(C_{e}>R_{b}-R_{s}\right)\\ \; & = & \;Pr\left(\gamma_{e}>2^{2(R_{b}-R_{s})}-1\right)\\ \; & = & \;Pr\left({∥g∥}^{2}>\frac{∥{\hat{h}}_{1}{∥}^{2}\left(2^{2\left(R_{b}-R_{s}\right)}-1\right)\sigma_{n}^{2}/\sigma_{u}^{2}}{{\left(\alpha -\left(1-\alpha \right)\left(2^{2\left(R_{b}-R_{s}\right)}-1\right)\right)}^{+}\sigma_{u}^{2}}\right)\\ \; & = & \;\frac{\int_{0}^{\infty }x^{N_{A}-1}e^{-\frac{x}{\left(1-\beta \right)\sigma_{h}^{2}}}\Gamma \left(N_{E},\Phi x\right)dx}{{\left(\left(1-\beta \right)\sigma_{h}^{2}\right)}^{N_{A}}\Gamma \left(N_{A}\right)}\\ \; & = & \;\frac{\Gamma (N_{A}+N_{E})}{N_{A}\Gamma \left(N_{A}\right)}\frac{{\left(\Phi \left(1-\beta \right)\sigma_{h}^{2}\right)}^{N_{E}}}{{\left(1+\Phi \left(1-\beta \right)\sigma_{h}^{2}\right)}^{N_{A}+N_{E}}}{}_{2}F_{1}\left(1,N_{A}+N_{E};N_{A}+1;\frac{1}{1+\Phi \left(1-\beta \right)\sigma_{h}^{2}}\right), \end{array}$$
where
(16)$$\Phi =\frac{\left(2^{2\left(R_{b}-R_{s}\right)}-1\right)\sigma_{n}^{2}/\left(\sigma_{u}^{2}\sigma_{g}^{2}\right)}{{\left(\alpha -\left(1-\alpha \right)\left(2^{2\left(R_{b}-R_{s}\right)}-1\right)\right)}^{+}},$$
${\left(.\right)}^{+}≜\max\left(0,.\right)$ and ${}_{2}F_{1}\left(\cdot \right)$ denotes the hypergeometric function [20].

*Special case:* When Eve has an unlimited number of antennas ($N_{E}\to \infty$) or is located close to Alice ($\sigma_{g}^{2}\to \infty$), we obtain from (Equation 15) that the secrecy outage probability converges to
(17)$$P_{so}\left({\hat{h}}_{1}\right)\to 1$$
if $\alpha -\left(1-\alpha \right)\left(2^{2\left(R_{b}-R_{s}\right)}-1\right)>0$, and, otherwise, is equal to 0 regardless of transmit power *P* and channel gain $\left|\right|{\hat{h}}_{1}{\left|\right|}^{2}$. This indicates that the secrecy is guaranteed only if the pair of codeword rates $\left(R_{b},R_{s}\right)$ is designed such that $\alpha -\left(1-\alpha \right)\left(2^{2\left(R_{b}-R_{s}\right)}-1\right)\leq 0$, or, equivalently,
(18)$$R_{s}\leq R_{b}-\frac{1}{2}\log_{2}\left(1+\frac{\alpha }{1-\alpha }\right),$$
which yields the zero secrecy outage probability regardless of Eve’s number of antennas and location. It is also noted that the equality of (Equation 18) is called the zero-outage secrecy rate.

## 4. Connection Outage Probability

Since the AN is received in both phases, Bob can remove it by simply processing $y_{b,1}-\sqrt{1-\alpha }y_{b,2}$ to obtain
(19)$$\begin{array}{ccc} y_{b}\; & = & \;y_{b,1}-\sqrt{1-\alpha }y_{b,2}\\ \; & = & \;\sqrt{\alpha }u+\frac{{\tilde{h}}_{1}^{T}{\hat{h}}_{1}^{*}}{∥{\hat{h}}_{1}{∥}^{2}}\left(\sqrt{\alpha }u+\sqrt{1-\alpha }v\right)-\frac{{\tilde{h}}_{2}^{T}{\hat{h}}_{2}^{*}}{∥{\hat{h}}_{2}{∥}^{2}}\sqrt{1-\alpha }v+\left(n_{b,1}-\sqrt{1-\alpha }n_{b,2}\right). \end{array}$$
Then, the SINR at Bob, viewing the second and third terms of (Equation 19) as noise, can be obtained by [17]
(20)$$\gamma_{b}=\frac{\alpha \sigma_{u}^{2}}{{\left|\frac{{\tilde{h}}_{1}^{T}{\hat{h}}_{1}^{*}}{∥{\hat{h}}_{1}∥}\right|}^{2}\frac{\sigma_{u}^{2}}{∥{\hat{h}}_{1}{∥}^{2}}+{\left|\frac{{\tilde{h}}_{2}^{T}{\hat{h}}_{2}^{*}}{∥{\hat{h}}_{2}∥}\right|}^{2}\frac{\left(1-\alpha \right)\sigma_{u}^{2}}{∥{\hat{h}}_{2}{∥}^{2}}+\left(2-\alpha \right)\sigma_{n}^{2}},$$
and then, the capacity at Bob is given by
(21)$$\begin{array}{c} C_{b}=\frac{1}{2}\log_{2}\left(1+\gamma_{b}\right), \end{array}$$
where the 1/2 multiplier represents the two-phase operation. Under a fixed codeword rate $R_{b}$, an outage event happens when the channel capacity at Bob falls below the target rate $R_{b}$ [18]. The probability of connection outage can be obtained from (Equation 20) and (Equation 21) as
(22)$$\begin{array}{ccc} P_{co}\; & = & \;Pr\left(C_{b}<R_{b}\right)\\ \; & = & \;Pr\left({\left|\frac{{\tilde{h}}_{1}^{T}{\hat{h}}_{1}^{*}}{∥{\hat{h}}_{1}∥}\right|}^{2}\frac{1}{∥{\hat{h}}_{1}{∥}^{2}}>k-{\left|\frac{{\tilde{h}}_{2}^{T}{\hat{h}}_{2}^{*}}{∥{\hat{h}}_{2}∥}\right|}^{2}\frac{\left(1-\alpha \right)}{∥{\hat{h}}_{2}{∥}^{2}}\right) \end{array}$$
Since it follows from Appendix A that the cumulative density function (CDF) and probability density function (PDF) of $|{\tilde{h}}_{i}^{T}{\hat{h}}_{i}^{*}/∥{\hat{h}}_{i}{{∥}^{2}|}^{2}$ is given by
(23)$$\begin{array}{ccc} F(x)\; & = & \;1-\frac{1}{{\left(1+x(1-\beta )/\beta \right)}^{N_{A}}},\\ f(x)\; & = & \;\frac{N_{A}\left(1-\beta \right)/\beta }{{\left(1+x(1-\beta )/\beta \right)}^{N_{A}+1}}, \end{array}$$
we can obtain the connection outage probability as
(24)$$\begin{array}{ccc} P_{co}\; & = & \;1-\int_{0}^{k/(1-\alpha )}F(k-x(1-\alpha ))f(x)dx,\\ \; & = & \;\int_{0}^{\frac{k}{1-\alpha }}\;\frac{N_{A}\left(1-\beta \right)dx/\beta }{{\left(1+\left(k-x\left(1-\alpha \right)\right)\frac{1-\beta }{\beta }\right)}^{N_{A}+1}{\left(1+x\frac{1-\beta }{\beta }\right)}^{N_{A}}} \end{array}$$
(25)$$\begin{array}{c} +\frac{1}{{\left(1+k(1-\beta )/((1-\alpha )\beta )\right)}^{N_{A}}}, \end{array}$$
where
(26)$$k=\frac{\alpha }{2^{2R_{b}}-1}-\frac{\left(2-\alpha \right)\sigma_{n}^{2}}{\sigma_{u}^{2}}.$$

For $P_{co}\leq \delta$, where δ indicates the reliability constraint, it can be obtained from (Equation 25) that
(27)$$R_{b}\leq R_{b}^{*},$$
and $R_{b}^{*}$ can be obtained by numerically searching for $R_{b}$, which results in $P_{co}\left(R_{b}\right)=\delta$. It should be noted that the exhaustive search is offline and the data value of $R_{b}^{*}$ will be stored according to the system parameters. Therefore, it is not being computed online, which would yield latency in real wireless systems.

## 5. Zero-Outage Secrecy Throughput and Energy Efficiency

The zero-outage secrecy throughput, denoted by η, is defined as the amount of information securely received at Bob subject to zero secrecy outage and quality of service, which is
(28)$$\begin{array}{ccc} \eta \; & = & \;\max_{R_{s}}\;R_{s}\times \left(1-\delta \right)\\ \; & & \;s.t.\;P_{co}\leq \delta \\ \; & & \;P_{so}\to 0. \end{array}$$
It follows from (Equation 18) and (Equation 27) that the zero-outage secrecy throughput can be found as
(29)$$\begin{array}{ccc} \eta \; & = & \;{\left[R_{b}^{*}-\frac{1}{2}\log_{2}\left(1+\frac{\alpha }{1-\alpha }\right)\right]}^{+}\times \left(1-\delta \right). \end{array}$$
Then, the corresponding energy efficiency of the system is defined as the zero-outage secrecy spectral efficiency per power consumption [21], which is
(30)$$\begin{array}{ccc} \zeta \; & = & \;\frac{B\times \eta }{(N_{A}P_{A}+P_{B})\times 2+P/\mu }, \end{array}$$
where $P_{A}$ and $P_{B}$ are the power consumption of each transmitter and receiver antenna, respectively, *B* is the bandwidth, and μ is the power amplifier coefficient. It should be noted that double circuit power in the denominator of (Equation 30) indicates two-phase transmissions.

## 6. Numerical Results and Discussion

In this section, we provide the numerical results of the analytical sections. Consider the GSM-1900 standard for a micro-cell model, which defines [21,22]: circuit powers with $P_{A}=0.36\;W$ and $P_{B}=0.24\;W$, noise power of $\sigma_{n}^{2}=N_{f}N_{0}B$ with $N_{0}=-174$ dBm/Hz, $N_{f}=3$ dB, and bandwidth B=200 kHz, and channel variance $\sigma_{h}^{2}=10^{-(3.45+3.8\log_{10}\left(d_{B}\right))}$, with the distance $d_{B}=1$ km between Alice and Bob.

From Figure 2 and Figure 3, one can see that zero-outage secrecy can always be achieved when the secrecy and codeword rate, i.e., $R_{s}$ and $R_{b}$, are designed properly to force the equivocate rate ($R_{b}-R_{s}$) to be larger than a threshold, as in (Equation 18). One can also see that the secrecy outage probability increases with increasing Eve’s number of antennas and with a closer distance to Alice. These results show that the noise injection scheme can help to guarantee secrecy with probability 1, which is crucial in real scenarios to prevent thieves from eavesdropping on credit card numbers under any circumstance. Furthermore, the secrecy outage probability is shown to increase with an increasing number of the transmitter antennas, as illustrated in Figure 4.

Figure 5 and Figure 6 depict the zero-outage secrecy throughput of the system versus α and *P* (dB), respectively. One can see that there exists an optimum ratio between the secret message and AN powers to maximize throughput, and the optimal power ratio increases when the transmit power increases. One can also see that when optimal α is applied, the secrecy throughput converges to a constant. This convergence of the secrecy throughput is due to the interference from estimation error.

Figure 7 and Figure 8 depict the SEE versus the average total transmit power *P* (dB) for different numbers of transmit antennas, $N_{A}$, and estimation error coefficient, β, at Alice. There exists an optimal *P* to maximize energy efficiency and that energy efficiency increases as the estimation error decreases. Furthermore, there also exists an optimal $N_{A}$ to maximize energy efficiency, as in Figure 9, depicting the SEE versus Alice’s numbers of transmit antennas for different transmit powers *P* (dB). This is because increasing the transmit power or number of transmit antennas significantly increases the total power consumption while the secrecy throughput stays constant (as shown in Figure 6), hence reducing the energy efficiency.

## 7. Conclusions

A new artificial noise (AN) injection method for MISOME systems was proposed. We proved that zero-outage secrecy can be always achieved in any circumstance. The numerical results showed the scheme can achieve zero secrecy outage with a trade-off in the secrecy rate and half-time transmissions. The results also showed that with perfect CSI, the zero-outage secrecy throughput increases with the transmit power, which is important in secrecy design. The results also showed the existence of an optimal transmit power and the number of transmitter antennas to maximize energy efficiency. As an extension of this work, antenna selection to reduce the complexity as well as increase energy efficiency will be considered. Also, this type of transmission strategy can be applied to multi-user scenarios, providing zero secrecy outage to the whole network with one simple artificial noise injection transmission.

## Figures and Tables

**Figure 1 entropy-25-01594-f001:**
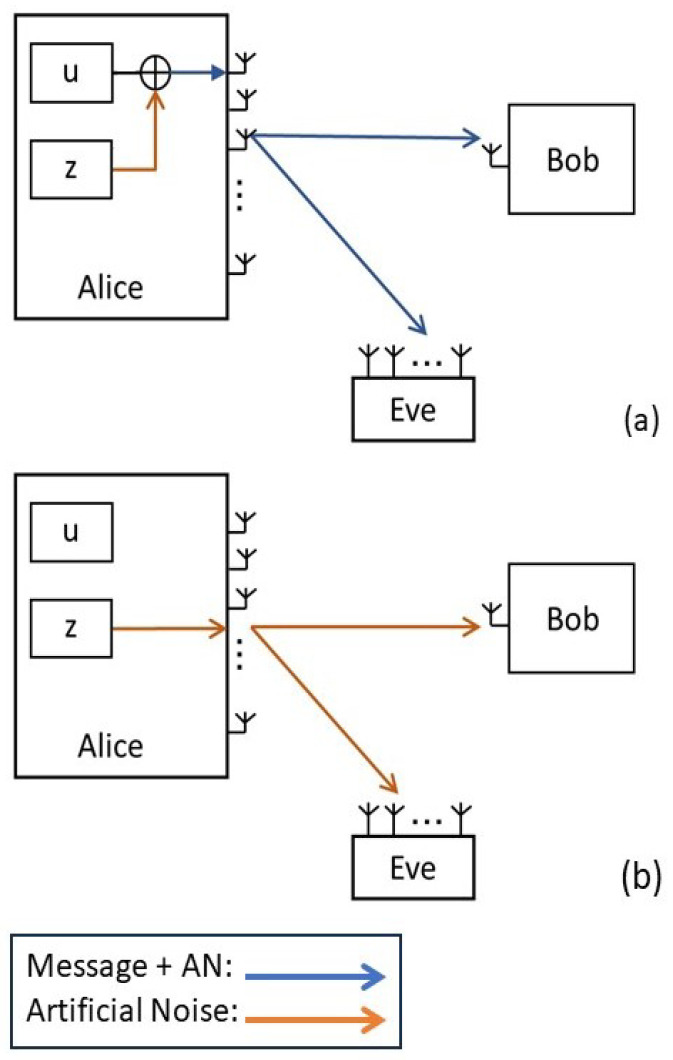
Two-phase transmission model: (**a**) Alice transmits noise-injected message in phase 1. (**b**) Alice transmits only AN in phase 2.

**Figure 2 entropy-25-01594-f002:**
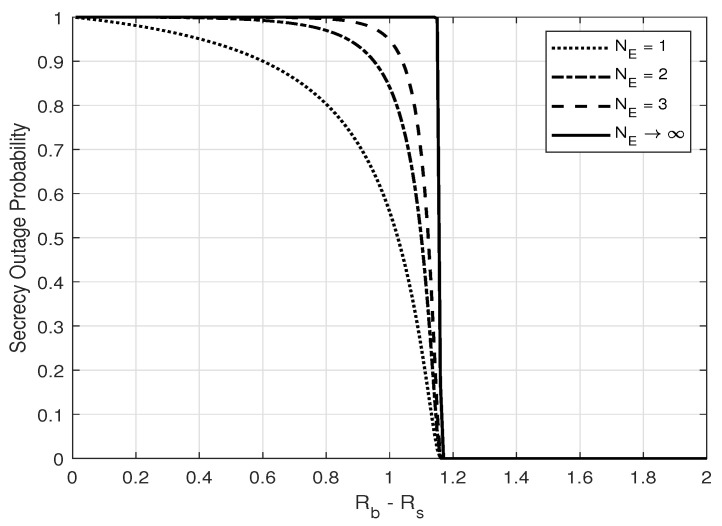
Secrecy outage probability, $P_{SO}$, versus $R_{b}-R_{s}$ for different values of $N_{E}$; α=0.8, $N_{A}=2$, and $\sigma_{g}^{2}=\sigma_{h}^{2}$.

**Figure 3 entropy-25-01594-f003:**
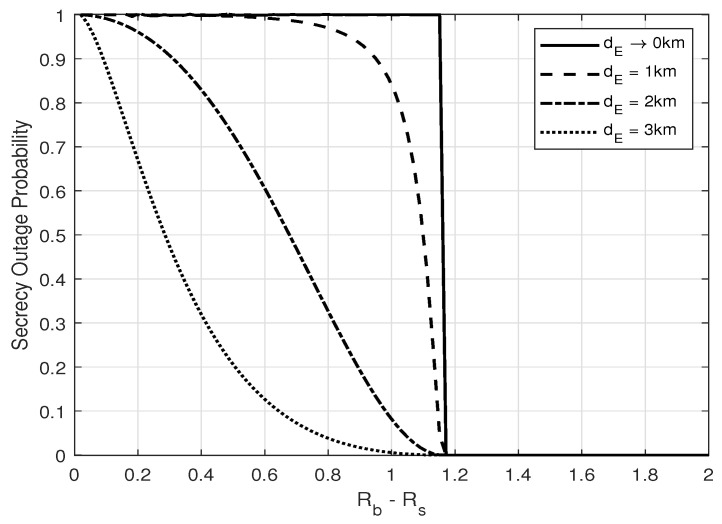
Secrecy outage probability, $P_{SO}$, versus $R_{b}-R_{s}$ for different eavesdropper distance $d_{E}$; α=0.8, $N_{A}=2$, and $N_{E}=2$.

**Figure 4 entropy-25-01594-f004:**
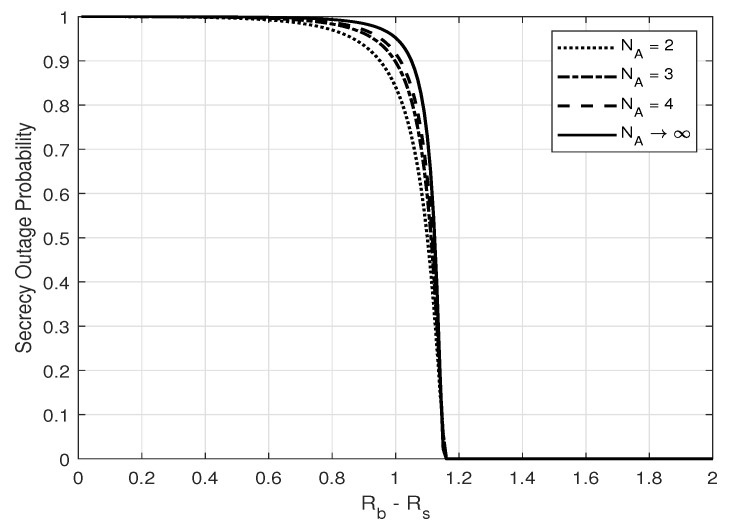
Secrecy outage probability, $P_{SO}$, versus $R_{b}-R_{s}$ for different values of $N_{A}$; α=0.8, $N_{E}=2$, and $\sigma_{g}^{2}=\sigma_{h}^{2}$.

**Figure 5 entropy-25-01594-f005:**
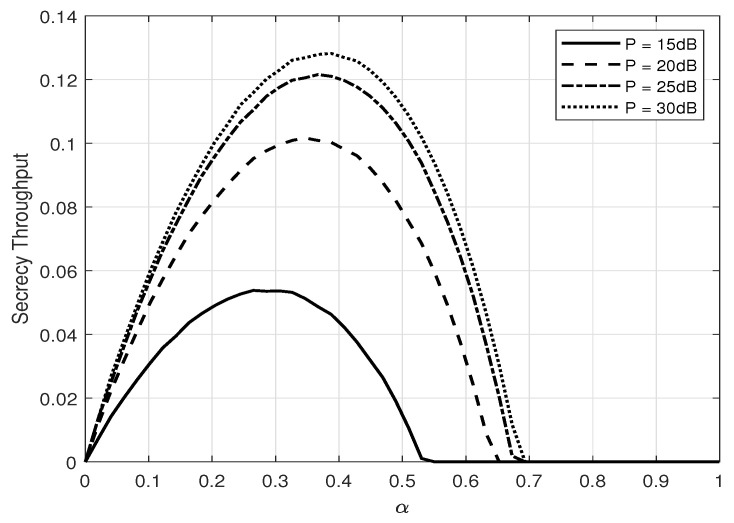
Secrecy throughput, η, versus α for different number of transmit power *P* (dB); β=0.1, δ=0.1, and $N_{A}=2$.

**Figure 6 entropy-25-01594-f006:**
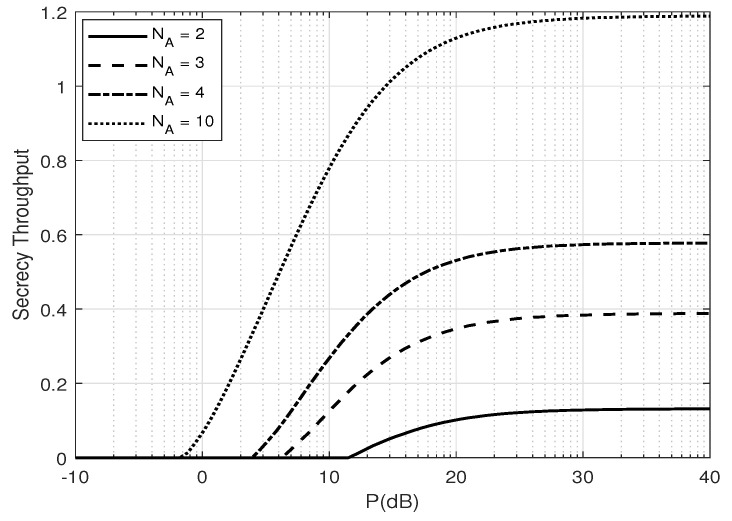
Secrecy throughput, η, versus *P* when the optimal α is applied for a different number of transmit antennas $N_{A}$; β=0.1 and δ=0.1.

**Figure 7 entropy-25-01594-f007:**
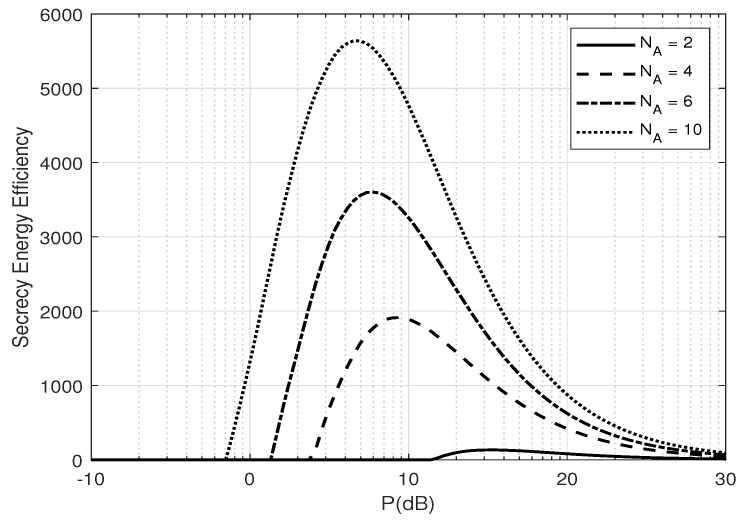
Secrecy energy efficiency, ζ, versus *P* for a different number of transmit antennas $N_{A}$ when optimal α is used; β=0.1 and δ=0.1.

**Figure 8 entropy-25-01594-f008:**
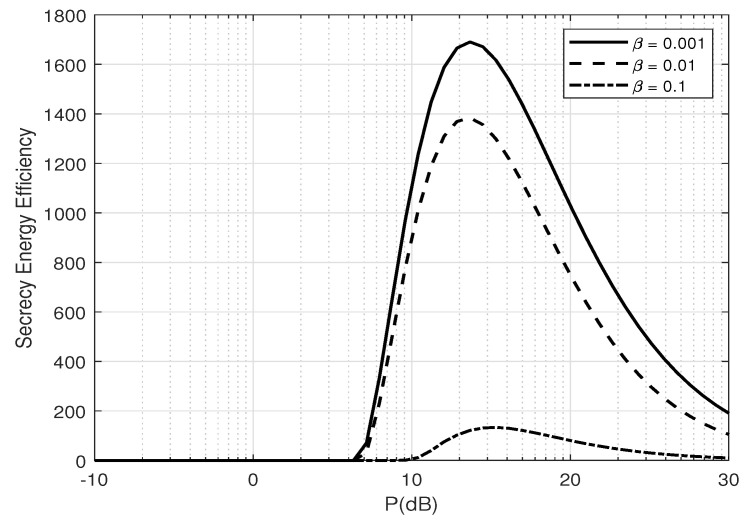
Secrecy energy efficiency, ζ, versus *P* for different number of β when the optimal α is applied; $N_{A}=2$, $N_{E}=2$, and δ=0.1.

**Figure 9 entropy-25-01594-f009:**
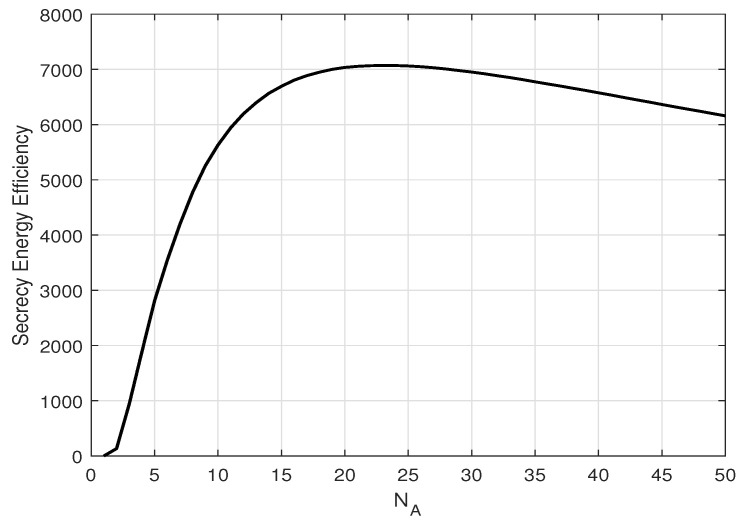
Secrecy energy efficiency, ζ, versus $N_{A}$ when optimal α and optimal power *P* is used; β=0.1 and δ=0.1.

## Data Availability

The data presented in this study are available on request from the corresponding author.

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
