# Peer review of "Novel Noise Injection Scheme to Guarantee Zero Secrecy Outage under Imperfect CSI"

_entropy, 2023, doi:10.3390/e25121594_

Round 1

Reviewer 1 Report

Comments and Suggestions for Authors

The paper proposes an artificial noise injection strategy under imperfect channel estimation at the legitimate channel to achieve zero secrecy outage probability under any circumstance. The contribution of this paper is trivial.

1. The AN scheme has been applied widely in existing literature. The topic of this paper is old, as can be seen from the references that are all published years ago.

2. The motivation as well as contribution of this paper are not well introduced.

3. The AN scheme is straightforward and simple, and there is no difficulty in deriving the probabilities under this scheme by following the routines in existing literature.

4. The proposed scheme is not compared with the state-of-the-art scheme in existing literature.

Author Response

Thank you very much for your review of this manuscript. We appreciate the time and effort you spend on the reviews. Please find the detailed responses and the authors hope that the response addresses all your concerns.

Reviewer 2 Report

Comments and Suggestions for Authors

This paper discusses a new strategy for injecting artificial noise into communication systems to ensure zero secrecy outage, even with imperfect channel estimation. It highlights the effectiveness of this approach in various scenarios, including when the eavesdropper has multiple antennas, and it provides insights into the impact of transmit power on secrecy throughput and energy efficiency. The paper is well-written and the novelty is good. I have some comments as follows:

  1. In the imperfection model for CSI given in (1), the error is modeled as a Gaussian distribution. What would happen if another imperfection model is considered, such as when the norm of the error is less than a threshold? In this case, we have h = h_hat + e, where |e| < epsilon.

  2. In this paper, Alice simply sends data to Bob. Why is beamforming not considered in this work?

  3. As a future work, the author could consider the impact of antenna selection and discuss what would happen if Alice performs antenna selection before transmitting data. This idea can also be included in the conclusion as a potential area for future research.

Comments on the Quality of English Language

The paper is well-written and easy to follow. However, there are some minor grammatical issues, so I suggest that the reader go over the paper one more time before the final submission.

Author Response

(The authors gave the same response as above.)

Reviewer 3 Report

Comments and Suggestions for Authors

The paper proposes a novel artificial noise (AN) injection strategy in MISOME systems under imperfect channel estimation at the legitimate channel to achieve zero secrecy outage probability under any circumstance.

It claims that zero secrecy outage is always achievable, regardless of the eavesdropper's number of antennas or location when the secrecy and codeword rates are chosen properly.

The results show that when there is perfect channel state information, the zero-outage secrecy throughput increases with transmit power.

The paper should clarify the following points:

The proposed method involves multiple transmissions of the same message: one transmission comprises noise and the message, and the other consists of noise only. It remains unclear to this reviewer how throughput is calculated. Is throughput assessed over both transmissions or just one?

Additionally, how is the codeword agreed upon?

The following mistakes should not be rectified:

Line 92 mentions 'Willie'.

Abbreviations should be defined before their use, such as 'MISOME' in the abstract and 'MMSE' on line 78.

Comments on the Quality of English Language

The presentation of the paper can be improved. Maths can be explained further. 

Author Response

(The authors gave the same response as above.)

Reviewer 4 Report

Comments and Suggestions for Authors

Author Response

(The authors gave the same response as above.)

Round 2

Reviewer 1 Report

Comments and Suggestions for Authors

Thanks for trying to revise the paper according to the comments. However, the revision is still not satisfactory. Existing schemes with properly modification to suit the system model are not adopted to compare the proposed scheme, which leads to insufficient verification.

Comments on the Quality of English Language

fine

Author Response

Thank you very much for your review of this manuscript. We appreciate the time and effort you spend on the reviews. The authors try to address all concerns of reviewers and hope to receive further feedback to improve the paper.  

Reviewer 4 Report

Comments and Suggestions for Authors

1) The authors said in abstract that the considered system is MISOME (multiple eaves.), but there is only one eavesdropper in the system in fact. This is inconsistent.

2) In addition to the difference between the conventional AN and noise injection, the authors are encouraged to outline their technical strengths, or challenging aspects of obtaining the results they wanted.

Author Response

(The authors gave the same response as above.)

Round 3

Reviewer 1 Report

Comments and Suggestions for Authors

I have no further comment.

Comments on the Quality of English Language

none

Reviewer 4 Report

Comments and Suggestions for Authors

Thank you for your efforts on editing the manuscript.